# General Tensor Spectral Co-clustering for Higher-Order Data

**Tao Wu**
Purdue University
wu577@purdue.edu

**Austin R. Benson**
Stanford University
arbenson@stanford.edu

**David F. Gleich**
Purdue University
dgleich@purdue.edu

## Abstract

Spectral clustering and co-clustering are well-known techniques in data analysis, and recent work has extended spectral clustering to square, symmetric tensors and hypermatrices derived from a network. We develop a new tensor spectral co-clustering method that simultaneously clusters the rows, columns, and slices of a nonnegative three-mode tensor and generalizes to tensors with any number of modes. The algorithm is based on a new random walk model which we call the super-spacey random surfer. We show that our method out-performs state-of-the-art co-clustering methods on several synthetic datasets with ground truth clusters and then use the algorithm to analyze several real-world datasets.

## 1 Introduction

Clustering is a fundamental task in machine learning that aims to assign closely related entities to the same group. Traditional methods optimize some aggregate measure of the strength of pairwise relationships (e.g., similarities) between items. Spectral clustering is a particularly powerful technique for computing the clusters when the pairwise similarities are encoded into the adjacency matrix of a graph. However, many graph-like datasets are more naturally described by higher-order connections among several entities. For instance, multilayer or multiplex networks describe the interactions between several graphs simultaneously with node-node-layer relationships [17]. Nonnegative tensors are a common representation for many of these higher-order datasets. For instance the $i, j, k$ entry in a third-order tensor might represent the similarity between items $i$ and $j$ in layer $k$.

Here we develop the General Tensor Spectral Co-clustering (GTSC) framework for clustering tensor data. The algorithm takes as input a nonnegative tensor, which may be sparse, non-square, and asymmetric, and outputs subsets of indices from each dimension (co-clusters). Underlying our method is a new stochastic process that models higher-order Markov chains, which we call a *super-spacey random walk*. This is used to generalize ideas from spectral clustering based on random walks. We introduce a variant on the well-known conductance measure from spectral graph partitioning [24] that we call *biased conductance* and describe how this provides a tensor partition quality metric; this is akin to Chung's use of circulations to spectrally-partition directed graphs [7]. Essentially, biased conductance is the exit probability from a set following our new super-spacey random walk model.

We use experiments on both synthetic and real-world problems to validate the effectiveness of our method[1]. For the synthetic experiments, we devise a "planted cluster" model for tensors and show that GTSC has superior performance compared to other state-of-the-art clustering methods in recovering the planted clusters. In real-world tensor data experiments, we find that our GTSC framework identifies stop-words and semantically independent sets in $n$-gram tensors as well as worldwide and regional airlines and airports in a flight multiplex network.

## 1.1 Related work

The Tensor Spectral Clustering (TSC) algorithm [4], another generalization of spectral methods to higher-order graph data [4], is closely related. Both the perspective and high-level view are similar, but the details differ in important ways. For instance, TSC was designed for the case when the higher-order tensor recorded the occurrences of small subgraph patterns within the network. This imposes limitations, including how, because the tensor arose based on some underlying graph that the partitioning metric was designed explicitly for a graph. Thus, the applications are limited in scope and cannot model, for example, the airplane-airplane-airport multiplex network we analyze in Section 3.2. Second, for sparse data, the model used by TSC required a correction term with magnitude proportional to the sparsity in the tensor. In sparse tensors, this makes it difficult to accurately identify clusters, which we show in Section 3.1.

Most other approaches to tensor clustering proceed by using low-rank factorizations [15, 21]. or a $k$-means objective [16]. In contrast, our work is based on a stochastic interpretation (escape probabilities from a set), in the spirit of random walks in spectral clustering for graphs. There are also several methods specific to clustering multiplex networks [27, 20] and clustering graphs with multiple entities [11, 2]. Our method handles general tensor data, which includes these types of datasets as a special case. Hypergraphs clustering [14] can also model the higher-order structures of the data, and in the case of tensor data, it is approximated by a standard weighted graph.

## 1.2 Background on spectral clustering of graphs from the perspective of random walks

We first review graph clustering methods from the view of graph cuts and random walks, and then review the standard spectral clustering method using sweep cuts. In Section 2, we generalize these notions to higher-order data in order to develop our GTSC framework.

Let $A \in \mathbb{R}_+^{n \times n}$ be the adjacency matrix of an undirected graph $G = (V, E)$ and let $n = |V|$ be the number of nodes in the graph. Define the diagonal matrix of degrees of vertices in $V$ as $D = \mathrm{diag}(A\mathbf{e})$, where $\mathbf{e}$ is the vector with all ones. The graph Laplacian is $L = D - A$ and the transition matrix is $P = AD^{-1} = A^T D^{-1}$. The transition matrix represents the transition probabilities of a random walk on the graph. If a walker is at node $j$, it transitions to node $i$ with probability $P_{ij} = A_{ji}/D_{jj}$.

**Conductance.** One of the most widely-used quality metrics for partitioning a graph's vertices into two sets $S$ and $\bar{S} = V \backslash S$ is conductance [24]. Intuitively, conductance measures the ratio of the number of edges in the graph that go between $S$ and $\bar{S}$ to the number of edges in $S$ or $\bar{S}$. Formally, we define conductance as:

$$\phi(S) = \mathrm{cut}(S)/\min\big(\mathrm{vol}(S), \mathrm{vol}(\bar{S})\big), \tag{1}$$

where

$$\mathrm{cut}(S) = \sum_{i \in S, j \in \bar{S}} A_{ij} \qquad \text{and} \qquad \mathrm{vol}(S) = \sum_{i \in S, j \in V} A_{ij}. \tag{2}$$

A set $S$ with small conductance is a good partition $(S, \bar{S})$. The following well-known observation relates conductance to random walks on the graph.

**Observation 1 ([18])** *Let $G$ be undirected, connected, and not bipartite. Start a random walk $(Z_t)_{t \in \mathbb{N}}$ where the initial state $X_0$ is randomly chosen following the stationary distribution of the random walk. Then for any set $S \in V$,*

$$\phi(S) = \max\big\{\Pr(Z_1 \in \bar{S} \mid Z_0 \in S), \Pr(Z_1 \in S \mid Z_0 \in \bar{S})\big\}.$$

This provides an alternative view of conductance—it measures the probability that one step of a random walk will traverse between $S$ and $\bar{S}$. This random walk view, in concert with the super-spacey random walk, will serve as the basis for our biased conductance idea to partition tensors in Section 2.4.

**Partitioning with a sweep cut.** Finding the set of minimum conductance is an NP-hard combinatorial optimization problem [26]. However, there are real-valued relaxations of the problem that are tractable to solve and provide a guaranteed approximation [19, 9]. The most well known computes an eigenvector called the Fiedler vector and then uses a sweep cut to identify a partition based on this eigenvector.

The Fiedler eigenvector $\mathbf{z}$ solves $\boldsymbol{L}\mathbf{z} = \lambda \boldsymbol{D}\mathbf{z}$ where $\lambda$ is the second smallest generalized eigenvalue. This can be equivalently formulated in terms of the random walk transition matrix $\boldsymbol{P}$. Specifically,

$$\boldsymbol{L}\mathbf{z} = \lambda \boldsymbol{D}\mathbf{z} \quad \Leftrightarrow \quad (\boldsymbol{I} - \boldsymbol{D}^{-1}\boldsymbol{A})\mathbf{z} = \lambda \mathbf{z} \quad \Leftrightarrow \quad \mathbf{z}^T \boldsymbol{P} = (1 - \lambda)\mathbf{z}^T.$$

The *sweep cut procedure* to identify a low-conductance set $S$ from $\mathbf{z}$ is as follows:

1. Sort the vertices by $\mathbf{z}$ as $z_{\sigma_1} \leq z_{\sigma_2} \leq \cdots \leq z_{\sigma_n}$.
2. Consider the $n-1$ candidate sets $S_k = \{\sigma_1, \sigma_2, \cdots, \sigma_k\}$ for $1 \leq k \leq n-1$
3. Choose $S = \operatorname{argmin}_{S_k} \phi(S_k)$ as the solution set.

The solution set $S$ from this algorithm satisfies the celebrated Cheeger inequality [19, 8]: $\phi(S) \leq 2\sqrt{\phi_{opt}}$, where $\phi_{opt} = \min_{S \subset V} \phi(S)$ is the minimum conductance over any set of nodes. Computing $\phi(S_k)$ for all $k$ only takes time linear in the number of edges in the graph because $S_{k+1}$ and $S_k$ differ only in the vertex $\sigma_{k+1}$.

To summarize, the spectral method requires two components: the second left eigenvector of $\boldsymbol{P}$ and the conductance criterion. We generalize these ideas to tensors in the following section.

## 2 A higher-order spectral method for tensor co-clustering

We now generalize the ideas from spectral graph partitioning to nonnegative tensor data. We first review our notation for tensors and then show how tensor data can be interpreted as a higher-order Markov chain. We briefly review Tensor Spectral Clustering [4] before introducing the new super-spacey random walk that we use here. This super-spacey random walk will allow us to compute a vector akin to the Fiedler vector for a tensor and to generalize conductance to tensors. Furthermore, we generalize the ideas from co-clustering in bipartite graph data [10] to rectangular tensors.

### 2.1 Preliminaries and tensor notation

We use $\underline{\boldsymbol{T}}$ to denote a tensor. As a generalization of a matrix, $\underline{\boldsymbol{T}}$ has $m$ indices (making $\underline{\boldsymbol{T}}$ an $m$th-order or $m$-mode tensor), with the $(i_1, i_2, \cdot, i_m)$ entry denoted $\underline{T}_{i_1, i_2, \cdots, i_m}$. We will work with non-negative tensors where $\underline{T}_{i_1, i_2, \cdots, i_m} \geq 0$. We call a subset of the tensor entries with all but the first element fixed a *column* of the tensor. For instance, the $j, k$ column of a three-mode tensor $\underline{\boldsymbol{T}}$ is $\underline{T}_{:,j,k}$. A tensor is *square* if the dimension of all the modes is equal and rectangular if not, and a square tensor is *symmetric* if it is equal for any permutation of the indices. For simplicity in the remainder of our exposition, we will focus on three-mode tensors. However, all of or ideas generalize to an arbitrary number of modes. (See, e.g., the work of Gleich et al. [13] and Benson et al. [5] for representative examples of how these generalizations work.) Finally, we use two operations between a tensor and a vector. First, a tensor-vector product with a three-mode tensor can output a vector, which we denote by:

$$\mathbf{y} = \underline{\boldsymbol{T}}\mathbf{x}^2 \quad \Leftrightarrow \quad y_i = \sum_{j,k} \underline{T}_{i,j,k} x_j x_k.$$

Second, a tensor-vector product can also produce a matrix, which we denote by:

$$\boldsymbol{A} = \underline{\boldsymbol{T}}[\mathbf{x}] \quad \Leftrightarrow \quad A_{i,j} = \sum_k \underline{T}_{i,j,k} x_k.$$

### 2.2 Forming higher-order Markov chains from nonnegative tensor data

Recall from Section 1.2 that we can form the transition matrix for a Markov chain from a square non-negative matrix $\boldsymbol{A}$ by normalizing the columns of the matrix $\boldsymbol{A}^T$. We can generalize this idea to define a higher-order Markov chain by normalizing a square tensor. This leads to a probability transition tensor $\underline{\boldsymbol{P}}$:

$$\underline{P}_{i,j,k} = \underline{T}_{i,j,k} / \sum_i \underline{T}_{i,j,k} \tag{3}$$

where we assume $\sum_i \underline{T}_{i,j,k} > 0$. In Section 2.3, we will discuss the sparse case where the column $\underline{T}_{:,j,k}$ may be entirely zero. When that case does not arise, entries of $\underline{\boldsymbol{P}}$ can be interpreted as the transition probabilities of a second-order Markov chain $(Z_t)_{t \in \mathbb{N}}$:

$$\underline{P}_{i,j,k} = \Pr(Z_{t+1} = i \mid Z_t = j, Z_{t-1} = k).$$

In other words, If the last two states were $j$ and $k$, then the next state is $i$ with probability $\underline{P}_{i,j,k}$.

It is possible to turn any higher-order Markov chain into a first-order Markov chain on the product state space of all ordered pairs $(i, j)$. The new Markov chain moves to the state-pair $(i, j)$ from $(j, k)$ with probability $\underline{P}_{i,j,k}$. Computing the Fiedler vector associated with this chain would be one approach to tensor clustering. However, there are two immediate problems. First, the eigenvector is of size $n^2$, which quickly becomes infeasible to store. Second, the eigenvector gives information about the product space—not the original state space. (In future work we plan to explore insights from marginals of this distribution.)

Recent work uses the *spacey random walk* and *spacey random surfer* stochastic processes to circumvent these issues [5]. The process is non-Markovian and generates a sequence of states $X_t$ as follows. After arriving at state $X_t$, the walker promptly "spaces out" and forgets the state $X_{t-1}$, yet it still wants to transition according to the higher-order transitions $\underline{P}$. Thus, it invents a state $Y_t$ by drawing a random state from its history and then transitions to state $X_{t+1}$ with probability $\underline{P}_{X_{t+1}, X_t, Y_t}$. We denote $\mathrm{Ind}\{\cdot\}$ as the indicator event and $H_t$ as the history of the process up to time $t$,[2] then

$$\Pr(Y_t = j \mid H_t) = \tfrac{1}{t+n}\left(1 + \sum_{r=1}^{t} \mathrm{Ind}\{X_r = j\}\right). \tag{4}$$

In this case, we assume that the process has a non-zero probability of picking any state by inflating its history count by 1 visit. The spacey random surfer is a generalization where the walk follows the above process with probability $\alpha$ and teleports at random following a stochastic vector $\mathbf{v}$ with probability $1 - \alpha$. This is akin to how the PageRank random walk includes teleportation.

Limiting stationary distributions are solutions to the multilinear PageRank problem [13]:

$$\alpha \underline{P}\mathbf{x}^2 + (1 - \alpha)\mathbf{v} = \mathbf{x}, \tag{5}$$

and the limiting distribution $\mathbf{x}$ represents the stationary distribution of the transition matrix $\underline{P}[\mathbf{x}]$ [5]. The transition matrix $\underline{P}[\mathbf{x}]$ asymptotically approximates the spacey walk or spacey random surfer.

Thus, it is feasible to compute an eigenvector of $\underline{P}[\mathbf{x}]$ matrix and use it with the sweep cut procedure on a generalized notion of conductance. However, this derivation assumes that all $n^2$ columns of $\underline{T}$ were non-zero, which does not occur in real-world datasets. The TSC method adjusted the tensor $\underline{T}$ and replaced any columns of all zeros with the uniform distribution vector [4]. Because the number of zero-columns may be large, this strategy dilutes the information in the eigenvector (see Appendix D.1). We deal with this issue more generally in the following section, and note that our new solution outperforms TSC in our experiments (Section 3).

## 2.3   A stochastic process for sparse tensors

Here we consider another model of the random surfer that avoids the issue of undefined transitions—which correspond to columns of $\underline{T}$ that are all zero—entirely. If the surfer attempts to use an undefined transition, then the surfer moves to a random state drawn from history. Formally, define the set of feasible states by

$$\mathcal{F} = \{(j, k) \mid \sum_i \underline{T}_{i,j,k} > 0\}. \tag{6}$$

Here, the set $\mathcal{F}$ denotes all the columns in $\underline{T}$ that are non-zero. The transition probabilities of our proposed stochastic process are given by

$$\Pr(X_{t+1} = i \mid X_t = j, H_t) \tag{7}$$
$$= (1 - \alpha)v_i + \alpha \sum_k \Pr(X_{t+1} = i \mid X_t = j, Y_t = k, H_t)\Pr(Y_t = k \mid H_t)$$

$$\Pr(X_{t+1} = i \mid X_t = j, Y_t = k, H_t) = \begin{cases} \underline{T}_{i,j,k}/\sum_i \underline{T}_{i,j,k} & (j, k) \in \mathcal{F} \\ \frac{1}{n+t}(1 + \sum_{r=1}^{t} \mathrm{Ind}\{X_r = i\}) & (j, k) \notin \mathcal{F}, \end{cases} \tag{8}$$

where $v_i$ is the teleportation probability. Again $Y_t$ is chosen according to Equation (4). We call this process the *super-spacey random surfer* because when the transitions are not defined it picks a random state from history.

This process is a (generalized) vertex-reinforced random walk [3]. Let $\underline{P}$ be the normalized tensor $\underline{P}_{i,j,k} = T_{i,j,k}/\sum_i T_{i,j,k}$ only for the columns in $\mathcal{F}$ and where all other entries are zero. Stationary distributions of the stochastic process must satisfy the following equation:

$$\alpha \underline{P}\mathbf{x}^2 + \alpha(1 - \|\underline{P}\mathbf{x}^2\|_1)\mathbf{x} + (1 - \alpha)\mathbf{v} = \mathbf{x}, \tag{9}$$

where $\mathbf{x}$ is a probability distribution vector (see Appendix A.1 for a proof). At least one solution vector $\mathbf{x}$ must exist, which follows directly from Brouwer's fixed-point theorem. Here we give a sufficient condition for it to be unique and easily computable.

**Theorem 2.1** *If $\alpha < 1/(2m-1)$ then there is a unique solution $\mathbf{x}$ to (9) for the general $m$-mode tensor. Furthermore, the iterative fixed point algorithm*

$$\mathbf{x}_{k+1} = \alpha \underline{\boldsymbol{P}} \mathbf{x}_k^2 + \alpha(1 - \|\underline{\boldsymbol{P}} \mathbf{x}_k^2\|_1)\mathbf{x}_k + (1 - \alpha)\mathbf{v}_k \tag{10}$$

*will converge at least linearly to this solution.*

This is a nonlinear setting and tighter convergence results are currently unknown, but these are unlikely to be tight on real-world data. For our experiments, we found that high values (e.g., $0.95$) of $\alpha$ do not impede convergence. We use $\alpha = 0.8$ for all our experiments.

In the following section, we show how to form a Markov chain from $\mathbf{x}$ and then develop our spectral clustering technique by operating on the corresponding transition matrix.

### 2.4 First-order Markov approximations and biased conductance for tensor partitions

From Observation 1 in Section 1.2, we know that conductance may be interpreted as the exit probability between two sets that form a partition of the nodes in the graph. In this section, we derive an equivalent first-order Markov chain from the stationary distribution of the *super-spacey random surfer*. If this Markov chain was guaranteed to be reversible, then we could apply the standard definitions of conductance and the Fiedler vector. This will not generally be the case, and so we introduce a biased conductance measure to partition this non-reversible Markov chain with respect to starting in the stationary distribution of the super-spacey random walk. We use the second largest, real-valued eigenvector of the Markov chain as an approximate Fiedler vector. Thus, we can use the sweep cut procedure described in Section 1.2 to identify the partition.

**Forming a first-order Markov chain approximation.** In the following derivation, we use the property of the two tensor-vector products that $\underline{\boldsymbol{P}}[\mathbf{x}]\mathbf{x} = \underline{\boldsymbol{P}}\mathbf{x}^2$. The stationary distribution $\mathbf{x}$ for the super-spacey random surfer is equivalently the stationary distribution of the Markov chain with transition matrix

$$\alpha\big(\underline{\boldsymbol{P}}[\mathbf{x}] + \mathbf{x}(\mathbf{e}^T - \mathbf{e}^T\underline{\boldsymbol{P}}[\mathbf{x}])\big) + (1 - \alpha)\mathbf{v}\mathbf{e}^T.$$

(Here we have used the fact that $\mathbf{x} \geq 0$ and $\mathbf{e}^T\mathbf{x} = 1$.) The above transition matrix denotes transitioning based on a first-order Markov chain with probability $\alpha$, and based on a fixed vector $\mathbf{v}$ with probability $1 - \alpha$. We introduce this following first-order Markov chain

$$\tilde{\boldsymbol{P}} = \underline{\boldsymbol{P}}[\mathbf{x}] + \mathbf{x}(\mathbf{e}^T - \mathbf{e}^T\underline{\boldsymbol{P}}[\mathbf{x}]),$$

which represents a useful (but crude) approximation of the higher-order structure in the data. First, we determine how often we visit states using the *super-spacey random surfer* to get a vector $\mathbf{x}$. Then the Markov chain $\tilde{\boldsymbol{P}}$ will tend to have a large probability of spending time in states where the higher-order information concentrates. This matrix represents a first-order Markov chain on which we can compute an eigenvector and run a sweep cut.

**Biased conductance.** Consider a random walk $(Z_t)_{t \in \mathbb{N}}$. We define the *biased conductance* $\phi_{\mathbf{p}}(S)$ of a set $S \subset \{1, \dots, n\}$ to be

$$\phi_{\mathbf{p}}(S) = \max\big\{\Pr(Z_1 \in \bar{S} \mid Z_0 \in S),\ \Pr(Z_1 \in S \mid Z_0 \in \bar{S})\big\},$$

where $Z_0$ is chosen according to a fixed distribution $\mathbf{p}$. Just as with the standard definition of conductance, we can interpret biased conductance as an escape probability. However, the initial state $Z_0$ is not chosen following the stationary distribution (as in the standard definition with a reversible chain) but following $\mathbf{p}$ instead. This is why we call it *biased conductance*. We apply this measure to $\tilde{\boldsymbol{P}}$ using $\mathbf{p} = \mathbf{x}$ (the stationary distribution of the super-spacey walk). This choice emphasizes the higher-order information. Our idea of biased conductance is equivalent to how Chung defines a conductance score for a directed graph [7].

We use the eigenvector of $\tilde{\boldsymbol{P}}$ with the second-largest real eigenvalue as an analogue of the Fiedler vector. If the chain were reversible, this would be exactly the Fiedler vector. When it is not, then the vector coordinates still encode indications of state clustering [25]; hence, this vector serves as a principled heuristic. It is important to note that although $\tilde{\boldsymbol{P}}$ is a dense matrix, we can implement the two operations we need with $\tilde{\boldsymbol{P}}$ in time and space that depends only on the number of non-zeros of the sparse tensor $\underline{\boldsymbol{P}}$ using standard iterative methods for eigenvalues of matrices (see Appendix B.1).

## 2.5 Handling rectangular tensor data

So far, we have only considered square, symmetric tensor data. However, tensor data are often rectangular. This is usually the case when the different modes represent different types of data. For example, in Section 3.2, we examine a tensor $\underline{T} \in \mathbb{R}^{p \times n \times n}$ of airline flight data, where $\underline{T}_{i,j,k}$ represents that there is a flight from airport $j$ to airport $k$ on airline $i$. Our approach is to embed the rectangular tensor into a larger square tensor and then symmetrize this tensor, using approaches developed by Ragnarsson and Van Loan [23]. After the embedding, we can run our algorithm to simultaneously cluster rows, columns, and slices of the tensor. This approach is similar in style to the symmetrization of bipartite graphs for co-clustering proposed by Dhillon [10].

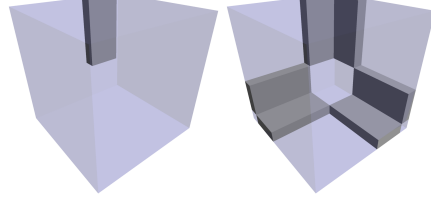

Let $\underline{U}$ be an $n$-by-$m$-by-$\ell$ rectangular tensor. Then we embed $\underline{U}$ into a square three-mode tensor $\underline{T}$ with $n + m + \ell$ dimensions and where $\underline{U}_{i,j,k} = \underline{T}_{i,j+n,k+n+m}$. This is illustrated in Figure 1 (left). Then we symmetrize the tensor by using all permutations of the indices Figure 1 (right). When viewed as a 3-by-3-by-3 block tensor, the tensor is

Figure 1: The tensor is first embedded into a larger square tensor (left) and then this square tensor is symmetrized (right).

$$\underline{T} = \left[ \begin{array}{ccc|ccc|ccc} 0 & 0 & 0 & 0 & 0 & \underline{U}_{(1,3,2)} & 0 & \underline{U}_{(1,2,3)} & 0 \\ 0 & 0 & \underline{U}_{(2,3,1)} & 0 & 0 & 0 & \underline{U}_{(2,1,3)} & 0 & 0 \\ 0 & \underline{U}_{(3,2,1)} & 0 & \underline{U}_{(3,1,2)} & 0 & 0 & 0 & 0 & 0 \end{array} \right],$$

where $\underline{U}_{(1,3,2)}$ is a generalized transpose of $\underline{U}$ with the dimensions permuted.

## 2.6 Summary of the algorithm

Our GTSC algorithm works by recursively applying the sweep cut procedure, similar to the recursive bisection procedures for clustering matrix-based data [6]. Formally for each cut, we:

1. Compute the super-spacey stationary vector $\mathbf{x}$ (Equation (9)) and form $\underline{P}[\mathbf{x}]$.
2. Compute second largest left, real-valued eigenvector $\mathbf{z}$ of $\tilde{P} = \underline{P}[\mathbf{x}] + \mathbf{x}(\mathbf{e}^T - \mathbf{e}^T \underline{P}[\mathbf{x}])$.
3. Sort the vertices by the eigenvector $\mathbf{z}$ as $z_{\sigma_1} \leq z_{\sigma_2} \leq \cdots \leq z_{\sigma_n}$.
4. Find the set $S_k = \{\sigma_1, \ldots, \sigma_k\}$ for which the biased conductance $\phi_{\mathbf{x}}(S_k)$ on transition matrix $\tilde{P}$ is minimized.

We continue partitioning as long as the clusters are large enough or we can get good enough splits. Specifically, if a cluster has dimension less than a specified size minimum size, we do not consider it for splitting. Otherwise, the algorithm recursively splits the cluster if either (1) its dimension is above some threshold or (2) the biased conductance of a new split is less than a target value $\phi^*$[3]. The overall algorithm is summarized in Appendix B as well as the algorithm complexity. Essentially, the algorithm scales linearly in the number of non-zeros of the tensor for each cluster that is produced.

# 3 Experiments

We now demonstrate the efficacy of our method by clustering synthetic and real-world data. We find that our method is better at recovering planted cluster structure in synthetically generated tensor data compared to other state-of-the-art methods. Please refer to Appendix C for the parameter details.

## 3.1 Synthetic data

We generate tensors with planted clusters and try to recover the clusters. For each dataset, we generate 20 groups of nodes that will serve as our planted clusters, where the number of nodes in each group from a truncated normal distribution with mean 20 and variance 5 so that each group has at least 4 nodes. For each group $g$ we also assign a weight $w_g$ where the weight depends on the group number. For group $i$, the weight is $(\sigma\sqrt{2\pi})^{-1} \exp(-(i - 10.5)^2/(2\sigma^2))$, where $\sigma$ varies by experiment. Non-zeros correspond to interactions between three indices (triples). We generate $t_w$ triples whose indices are within a group and $t_a$ triples whose indices span across more than one group.

Table 1: Adjusted Rand Index (ARI), Normalized Mutual Information (NMI), and F1 scores on various clustering methods for recovering synthetically generated tensor data with planted cluster structure. The $\pm$ entries are the standard deviation over 5 trials.

| | ARI | NMI | F1 | ARI | NMI | F1 |
|---|---|---|---|---|---|---|
| | Square tensor with $\sigma = 4$ | | | Rectangular tensor with $\sigma = 4$ | | |
| GTSC | **0.99**±0.01 | **0.99**±0.00 | **0.99**±0.01 | **0.97**±0.06 | **0.98**±0.03 | **0.97**±0.05 |
| TSC | 0.42±0.05 | 0.60±0.04 | 0.45±0.04 | 0.38±0.17 | 0.53±0.15 | 0.41±0.16 |
| PARAFAC | 0.82±0.05 | 0.94±0.02 | 0.83±0.04 | 0.81±0.04 | 0.90±0.02 | 0.82±0.04 |
| SC | **0.99**±0.01 | **0.99**±0.01 | **0.99**±0.01 | 0.91±0.06 | 0.94±0.04 | 0.91±0.06 |
| MulDec | 0.48±0.05 | 0.66±0.03 | 0.51±0.05 | 0.27±0.06 | 0.39±0.05 | 0.32±0.05 |
| | Square tensor with $\sigma = 2$ | | | Rectangular tensor with $\sigma = 2$ | | |
| GTSC | **0.78**±0.13 | **0.89**±0.06 | **0.79**±0.12 | **0.96**±0.06 | **0.97**±0.04 | **0.96**±0.06 |
| TSC | 0.41±0.11 | 0.60±0.09 | 0.44±0.10 | 0.28±0.08 | 0.44±0.10 | 0.32±0.08 |
| PARAFAC | 0.48±0.08 | 0.67±0.04 | 0.50±0.07 | 0.10±0.04 | 0.24±0.05 | 0.15±0.04 |
| SC | 0.43±0.07 | 0.66±0.04 | 0.47±0.06 | 0.38±0.07 | 0.52±0.05 | 0.41±0.07 |
| MulDec | 0.19±0.01 | 0.37±0.01 | 0.24±0.01 | 0.08±0.01 | 0.19±0.02 | 0.14±0.01 |

The $t_w$ triples are chosen by first uniformly selecting a group $g$ and then uniformly selecting three indices $i$, $j$, and $k$ from group $g$ and finally assigning a weight of $w_g$. For the $t_a$ triples, the sampling procedure first selects an index $i$ from group $g_i$ with a probability proportional to the weights of the group. In other words, indices in group $g$ are chosen proportional to $w_g$. Two indices $j$ and $k$ are then selected uniformly at random from groups $g_j$ and $g_k$ other than $g_i$. Finally, the weight in the tensor is assigned to be the average of the three group weights. For rectangular data, we follow a similar procedure where we distinguish between the indices for each mode of the tensor.

For our experiments, $t_w = 10,000$ and $t_a = 1,000$, and the variance $\sigma$ that controls the group weights is 2 or 4. For each value of $\sigma$, we create 5 sample datasets. The value of $\sigma$ affects the concentration of the weights and how certain groups of nodes interact with others. This skew reflects properties of the real-world networks we examine in the next section.

Our GTSC method is compared with Tensor Spectral Clustering (TSC) [4], the Tensor Decomposition PARAFAC [1], Spectral Clustering (SC) via Multilinear SVD [12] and Multilinear Decomposition (MulDec) [21]. Table 1 depicts the performances of the four algorithms in recovering the planted clusters. In all cases, GTSC has the best performance. We note that the running time is a few seconds for GTSC, TSC and SC and nearly 30 minutes for PARAFAC and MulDec per trial. Note that the tensors have roughly $50,000$ non-zeros. The poor scalability prohibits the later two methods from being applied to the real-world tensors in the following section.

## 3.2 Case study in airline flight networks

We now turn to studying real-world tensor datasets. We first cluster an airline-airport multimodal network which consists of global air flight routes from 539 airlines and $2,939$ airports[4]. In this application, the entry $\underline{T}_{i,j,k}$ of the three-mode tensor $\underline{T}$ is 1 if airline $i$ flies between airports $j$ and $k$ and 0 otherwise. Figure 2 illustrates the connectivity of the tensor with a random ordering of the indices (left) and the ordering given by the popularity of co-clusters (right). We can see that after the co-clustering, there is clear structure in the data tensor.

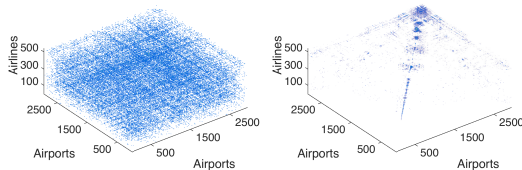

Figure 2: Visualization of the airline-airport data tensor. The $x$ and $y$ axes index airports and the $z$ axis indexes airlines. A dot represents that an airline flies between those two airports. On the left, indices are sorted randomly. On the right, indices are sorted by the co-clusters found by our GTSC framework, which reveals structure in the tensor.

One prominent cluster found by the method corresponds to large international airports in cities such as Beijing and New York City. This group only accounts for 8.5% of the total number of airports, but it is responsible for 59% of the total routes. Figure 2 illustrates this result—the airports with the highest

indices are connected to almost every other airport. This cluster is analogous to the "stop word" group we will see in the $n$-gram experiments. Most other clusters are organized geographically. Our GTSC framework finds large clusters for Europe, the United States, China/Taiwan, Oceania/SouthEast Asia, and Mexico/Americas. Interestingly, Cancún International Airport is included with the United States cluster, likely due to large amounts of tourism.

### 3.3 Case study on n-grams

Next, we study data from $n$-grams (consective sequences of words in texts). We construct a square mode-$n$ tensor where indices correspond to words. An entry in the tensor is the number of occurrences this $n$-gram. We form tensors from both English and Chinese corpora for $n = 3, 4$.[5] The non-zeros in the tensor consist of the frequencies of the one million most frequent $n$-grams.

**English n-grams.** We find several conclusions that hold for both tensor datasets. Two large groups in both datasets consist of stop words, i.e., frequently occuring connector words. In fact, 48% (3-gram) and 64% (4-gram) of words in one cluster are prepositions (e.g., *in, of, as, to*) and link verbs (e.g., *is, get, does*). In the another cluster, 64% (3-gram) and 57% (4-gram) of the words are pronouns (e.g., *we, you, them*) and link verbs. This result matches the structure of English language where link verbs can connect both prepositions and pronouns whereas prepositions and pronouns are unlikely to appear in close vicinity. Other groups consist of mostly semantically related English words, e.g.,

> {*cheese, cream, sour, low-fat, frosting, nonfat, fat-free*} and
> {*bag, plastic, garbage, grocery, trash, freezer*}.

The clustering of the 4-gram tensor contains some groups that the 3-gram tensor fails to find, e.g.,

> {*german, chancellor, angela, merkel, gerhard, schroeder, helmut, kohl*}.

In this case, Angela Merkel, Gerhard Schroeder, and Helmut Kohl have all been German chancellors, but it requires a 4-gram to make this connection strong. Likewise, some clusters only appear from clustering the 3-gram tensor. One such cluster is

> {*church, bishop, catholic, priest, greek, orthodox, methodist, roman, episcopal*}.

In 3-grams, we may see phrases such as "catholic church bishop", but 4-grams containing these words likely also contain stop words, *e.g.*, "bishop of the church". However, since stop words already form their own cluster, this connection is destoryed.

**Chinese n-grams.** We find that many of the conclusions from the English $n$-gram datasets also hold for the Chinese $n$-gram datasets. This includes groups of stop words and semantically related words. For example, there are two clusters consisting of mostly stop words (200 most frequently occurring words) from the 3-gram and 4-gram tensors. In the 4-gram data, one cluster of 31 words consists entirely of stop words and another cluster contains 36 total words, of which 23 are stop words.

There are some words from the two groups that are not typically considered as stop words, e.g.,

> 社会 *society,* 经济 *economy,* 发展 *develop,* 主义 *-ism,* 国家 *nation,* 政府 *government*

These words are also among the top 200 most common words according to the corpus. This is a consequence of the dataset coming from scanned Chinese-language books and is a known issue with the Google Books corpus [22]. In this case, it is a feature as we are illustrating the efficacy of our tensor clustering framework rather than making any linguistic claims.

## 4 CONCLUSION

In this paper we developed the General Tensor Spectral Co-clustering (GTSC) method for co-clustering the modes of nonnegative tensor data. Our method models higher-order data with a new stochastic process, the super-spacey random walk, which is a variant of a higher-order Markov chain. With the stationary distribution of this process, we can form a first-order Markov chain which captures properties of the higher-order data and then use tools from spectral graph partitioning to find co-clusters. In future work, we plan to create tensors that bridge information from multiple modes. For instance, clusters in the $n$-gram data depended on $n$, e.g., the names of various German chancellors only appeared as a 4-gram cluster. It would be useful to have a holistic tensor to jointly partition both 3- and 4-gram information.

**Acknowledgements.** TW and DFG are supported by NSF IIS-1422918 and DARPA SIMPLEX. ARB is supported by a Stanford Graduate Fellowship.

## Footnotes

[1]Code and data for this paper are available at: https://github.com/wutao27/GtensorSC

[2]Formally, this is the $\sigma$-algebra generated by the states $X_1, \dots, X_t$.

[3]We tested $\phi^*$ from 0.3 to 0.4, and we found the value of $\phi^*$ is not very sensitive to the experimental results.

[4]Data were collected from `http://openflights.org/data.html#route`.

[5]English $n$-gram data were collected from http://www.ngrams.info/intro.asp and Chinese $n$-gram data were collected from https://books.google.com/ngrams.

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
