[Supplementary Material]

# General Tensor Spectral Co-clustering
# for Higher-Order Data

**Tao Wu**
Purdue University
wu577@purdue.edu

**Austin R. Benson**
Stanford University
arbenson@stanford.edu

**David F. Gleich**
Purdue University
dgleich@purdue.edu

## A  Super-spacey random surfer

### A.1  Stationary distribution

Suppose the process has run for a very long time and that $\mathbf{x}_t$ is the current empirical distribution. From equation (7), we have

$$\Pr(X_{t+1} = i | X_t = j) = (1-\alpha)\mathbf{v}_i + \alpha \Big( \sum_{(j,k)\in\mathcal{F}} \underline{P}_{i,j,k}\mathbf{x}_t(k) + \sum_{(j,k)\notin\mathcal{F}} \mathbf{x}_t(k)\mathbf{x}_t(i) \Big). \tag{1}$$

The above transition process can be treated as a Markov chain with transition matrix depending on the current empirical distribution $\mathbf{x}_t$. Let $M_t$ denote this Markov chain transition matrix, and we have:

$$M_t = (1-\alpha)\mathbf{v}\mathbf{e}^T + \alpha\underline{P}[\mathbf{x}_t] + \alpha\mathbf{x}_t(\mathbf{e}^T - \mathbf{e}^T\underline{P}[\mathbf{x}_t]).$$

To understand the above expression, the first term of $M_t$ comes from the first term of equation (1) and so does the second term. For the third term, the $j$-th entry of $\mathbf{e}^T - \mathbf{e}^T\underline{P}[\mathbf{x}_t]$ is exactly $\sum_{(j,k)\notin\mathcal{F}} \mathbf{x}_t(k)$, so that $M_t$ is a column stochastic matrix. At stationarity of the super-spacey random walk, the stationary distribution of this Markov chain must satisfy $\mathbf{x} = M_t\mathbf{x}$. When the random walk is at its stationary distribution $\mathbf{x}$, $M_t$ will not change but be fixed as:

$$M_t = M = (1-\alpha)\mathbf{v}\mathbf{e}^T + \alpha\underline{P}[\mathbf{x}] + \alpha\mathbf{x}(\mathbf{e}^T - \mathbf{e}^T\underline{P}[\mathbf{x}]).$$

And we have:

$$\begin{aligned}
M\mathbf{x} &= (1-\alpha)\mathbf{v}\mathbf{e}^T\mathbf{x} + \alpha\underline{P}[\mathbf{x}]\mathbf{x} + \alpha\mathbf{x}(\mathbf{e}^T - \mathbf{e}^T\underline{P}[\mathbf{x}])\mathbf{x} \\
&= (1-\alpha)\mathbf{v} + \alpha\underline{P}\mathbf{x}^2 + \alpha\mathbf{x}(1 - \mathbf{e}^T\underline{P}\mathbf{x}^2) \\
&= (1-\alpha)\mathbf{v} + \alpha\underline{P}\mathbf{x}^2 + \alpha(1 - \|\underline{P}\mathbf{x}^2\|_1)\mathbf{x}.
\end{aligned}$$

Thus equation 9 gives us the necessary condition for $\mathbf{x}$ being the stationary distribution. A more formal proof is to use the results from [2] to show that (9) is the necessary condition of the stationary distribution in a vertex reinforced random walk.

### A.2  Proof of Theorem 2.1

Let $R$ denote the mode-1 unfolding of $\underline{P}$:

$$R = [\underline{P}(:,:,1) \mid \underline{P}(:,:,2) \mid \cdots \mid \underline{P}(:,:,n)].$$

Note that $R(\mathbf{x} \otimes \mathbf{x}) = \underline{P}\mathbf{x}^2$ where $\otimes$ is the Kronecker product. Assume $\mathbf{x}$ and $\mathbf{y}$ are two solutions of (9). Let $r_x = \|R(\mathbf{x} \otimes \mathbf{x})\|_1$ and $r_y = \|R(\mathbf{y} \otimes \mathbf{y})\|_1$. Then

$$\|\mathbf{x} - \mathbf{y}\|_1 \leq \alpha\|R(\mathbf{x} \otimes \mathbf{x} - \mathbf{y} \otimes \mathbf{y})\|_1 + \alpha\|(1-r_x)\mathbf{x} - (1-r_y)\mathbf{y}\|_1.$$

By Lemma 4.5 of [6], the first term

$$\alpha\|R(\mathbf{x} \otimes \mathbf{x} - \mathbf{y} \otimes \mathbf{y})\|_1 \leq \alpha\|R\|_1\|\mathbf{x} \otimes \mathbf{x} - \mathbf{y} \otimes \mathbf{y}\|_1 \leq 2\alpha\|\mathbf{x} - \mathbf{y}\|_1.$$

The second term satifies

$$
\begin{aligned}
& \alpha \|(1 - r_x)\mathbf{x} - (1 - r_y)\mathbf{y}\|_1 \\
& = \alpha \|(1 - r_x)(\mathbf{x} - \mathbf{y}) - (r_x - r_y)\mathbf{y}\|_1 \\
& \leq \alpha \|(1 - r_x)(\mathbf{x} - \mathbf{y})\|_1 + \alpha |r_y - r_x| \\
& \leq \alpha \|\mathbf{x} - \mathbf{y}\|_1 + \alpha \|\boldsymbol{R}(\mathbf{x} \otimes \mathbf{x} - \mathbf{y} \otimes \mathbf{y})\|_1 \leq 3\alpha \|\mathbf{x} - \mathbf{y}\|_1.
\end{aligned}
$$

Combining the above two facts, we know when $\alpha < 1/5$ the solution is unique. For an $m$-mode tensor, this idea generalizes to $\alpha < 1/(2m - 1)$.

For the convergence of the fixed point algorithm (10), the same analysis shows that $\|\mathbf{x}_{k+1} - \mathbf{x}^*\|_1 \leq 5\alpha \|\mathbf{x}_k - \mathbf{x}^*\|_1$, and so the iteration converges at least linearly when the solution is unique.

# B  Algorithm discussion

---
**Algorithm 1** General Tensor Spectral Co-clustering

---
**Require:**
  Symmetric square tensor $\underline{\boldsymbol{T}} \in \mathbb{R}_+^{n \times n \times n}$, $\alpha \in (0, 1)$
  Stopping criterion `max-size`, `min-size`, $\phi^*$
**Ensure:**
  Partitioning $C$ of indices $\{1, \dots, n\}$.
 1: $C = \{\{1, \dots, n\}\}$
 2: IF $n \leq$ `min-size`:   **RETURN**
 3: Generate transition tensor $\underline{\boldsymbol{P}}$ by

$$
\underline{P}_{ijk} = \begin{cases} \underline{T}_{ijk} / \sum_{i=1}^n \underline{T}_{ijk} & \text{if } \sum_{i=1}^n \underline{T}_{ijk} > 0 \\ 0 & \text{otherwise} \end{cases}
$$

 4: Compute super-spacey stationary vector $\mathbf{x}$ (Equation (9)) and form $\underline{\boldsymbol{P}}[\mathbf{x}]$.
 5: Compute second largest left, real-valued eigenvector $\mathbf{z}$ of
   $\tilde{\boldsymbol{P}} = \underline{\boldsymbol{P}}[\mathbf{x}] + \mathbf{x}(\mathbf{e}^T - \mathbf{e}^T \underline{\boldsymbol{P}}[\mathbf{x}])$ (that is, $\mathbf{z}^T \tilde{\boldsymbol{P}} = \lambda \mathbf{z}^T$).
 6: $\sigma \leftarrow$ Sort eigenvector $\mathbf{z}$
 7: $(S, \phi) \leftarrow$ Biased Conductance Sweep Cut$(\sigma, \underline{\boldsymbol{P}}[\mathbf{x}])$ with bias $\mathbf{p} = \mathbf{x}$.
 8: **if** $n \geq$ `max-size` or $\phi \leq \phi^*$ **then**
 9:    $C_S =$ Algorithm 1 on sub-tensor $\underline{\boldsymbol{T}}_{S,S,S}$.
10:    $C_{\bar{S}} =$ Algorithm 1 on sub-tensor $\underline{\boldsymbol{T}}_{\bar{S},\bar{S},\bar{S}}$.
11:    $C = C_S \cup C_{\bar{S}}$.
12: **end if**
13: **RETURN** $C$

---

The overall algorithm is summarized in Algorithm 1.

We also have a couple of pre-processing steps. First, we have to symmetrize the data if the tensor is rectangular. Second, we look for "empty" indices that do not participate in the tensor structure. Formally, index $i$ is empty if $\underline{\boldsymbol{T}}_{ijk} = 0$ for all $j$ and $k$.

## B.1  Linear time sweep cut

In this section we prove that the operations we need for computing the eigenvector and conducting the sweep cut for $\tilde{\boldsymbol{P}}$ (possibly a dense matrix) depend only on the number of non-zeros of the sparse tensor $\underline{\boldsymbol{P}}$.

For computing the eigenvector of $\tilde{\boldsymbol{P}}$, it only involves the mat-vec operation (i.e., $\tilde{\boldsymbol{P}}\mathbf{b}$):

$$
\tilde{\boldsymbol{P}}\mathbf{b} = \underline{\boldsymbol{P}}[\mathbf{x}]\mathbf{b} + \mathbf{x}\big(\mathbf{e}^T\mathbf{b} - \mathbf{e}^T(\underline{\boldsymbol{P}}[\mathbf{x}]\mathbf{b})\big)
$$

Since $\underline{\boldsymbol{P}}[\mathbf{x}]$ is a sparse matrix with number of non-zeros up to the number of non-zeros in $\underline{\boldsymbol{P}}$, so the mat-vec operation $\tilde{\boldsymbol{P}}\mathbf{b}$ also only depends on the number of non-zeros in $\underline{\boldsymbol{P}}$.

For the sweep cut procedure, let $\mathbf{z}$ be the eigenvector we computed from $\tilde{P}$ and without a loss of generality, we assume $z_1 \leq z_2 \leq \cdots \leq z_n$. We want to see what is the computational complexity for calculating $\Pr(X_1 \in \bar{S}_{k+1} | X_0 \in S_{k+1})$ when the value of $\Pr(X_1 \in \bar{S}_k | X_0 \in S_k)$ is given.

Denote the row vector $\mathbf{h} = \mathbf{e}^T - \mathbf{e}^T \mathbf{P}[\mathbf{x}]$, $H_k = \sum_{i=1}^{k} x_i h_i$ and $Q_k = \sum_{i=k}^{n} x_k$ Then given the value of:

$$\Pr(X_1 \in \bar{S}_k | X_0 \in S_k) = \frac{\Pr(X_1 \in \bar{S}_k, X_0 \in S_k)}{\Pr(X_0 \in S_k)} = \frac{p_k}{1 - Q_{k+1}}$$

To compute $\Pr(X_1 \in \bar{S}_{k+1} | X_0 \in S_{k+1})$

$$= \frac{\Pr(X_1 \in \bar{S}_{k+1}, X_0 \in S_k) + \Pr(X_1 \in \bar{S}_{k+1}, X_0 = k+1)}{\Pr(X_0 \in S_k) + \Pr(X_0 = k+1)}$$

$$= \frac{\Pr(X_1 \in \bar{S}_k, X_0 \in S_k) - \Pr(X_1 = k+1, X_0 \in S_k) + \Pr(X_1 \in \bar{S}_{k+1}, X_0 = k+1)}{1 - Q_{k+1} + x_{k+1}}$$

$$= \frac{p_k - \sum_{i=1}^{k} x_i \tilde{P}[\mathbf{x}]_{k+1,i} + x_{k+1} \sum_{i=k+2}^{n} \tilde{P}[\mathbf{x}]_{i,k+1}}{1 - Q_{k+2}}$$

$$= \frac{p_k - \sum_{i=1}^{k} x_i P[\mathbf{x}]_{k+1,i} - \sum_{i=1}^{k} x_i x_{k+1} h_i + x_{k+1} \sum_{i=k+2}^{n} P[\mathbf{x}]_{i,k+1} + x_{k+1} \sum_{i=k+2}^{n} x_i h_{k+1}}{1 - Q_{k+2}}$$

$$= \frac{- \sum_{i=1}^{k} x_i P[\mathbf{x}]_{k+1,i} + x_{k+1} \sum_{i=k+2}^{n} P[\mathbf{x}]_{i,k+1}}{1 - Q_{k+2}} + \frac{p_k - x_{k+1} H_k + x_{k+1} h_{k+1} Q_{k+2}}{1 - Q_{k+2}}$$

The first term only involves the $(k+1)$-th row and column of $P[\mathbf{x}]$, and the second term cost constant computation as long as the arraies $H$ and $Q$ are precomputed. Since computing $H$ and $Q$ is $O(n)$, the total complexity of computing the above probability for all $1 \leq k \leq n$ is linear to the number of non-zeros in $P[\mathbf{x}]$, and the same conclusion holds for $\Pr(X_1 \in \bar{S}_k | X_0 \in \bar{S}_k)$. So in summary the Sweep Cut procedure costs order of total number of non-zeros in $P[\mathbf{x}]$.

## B.2 Computational complexity

We now provide an analysis of the running time of our algorithm. Let $N$ be the number of non-zeros in the tensor $\underline{T}$. First, note that the pre-processing (tensor symmetrization and finding empty nodes) takes $O(N)$ time. Now, we examine the computational complexity of a single partition:

1. Generating the transition tensor $\underline{P}$ costs $O(N)$.
2. Each step of the fixed-point algorithm to find the stationary distribution is $O(N)$.
3. Constructing $\underline{P}[\mathbf{x}]$ costs $O(N)$. (The matrix $\tilde{P}$ is not formed explicitly).
4. Each iteration of the eigenvector computation takes time linear in the number of non-zeros in $\underline{P}[\mathbf{x}]$, which is $O(N)$.
5. Sorting the eigenvector takes $O(n \log n)$ computations, which is negligible considering $N$ is big compared to $n$.
6. The sweep cut takes time $O(n + N)$, which is $O(N)$.

In practice, we find that only a few iterations are needed to compute the stationary distribution, which is consistent with past results [3, 6]. For these systems, we do not know how many iterations are needed for the eigenvector computations. However, for the datasets we analyze in this paper, the eigenvector computation is not prohibitive. Thus, we can think of the time for each cut as roughly linear in the number of non-zeros. Provided that the cuts are roughly balanced, the depth of the recursion tree is $O(\log N)$, and the total time is $O(N \log N)$. Again, in our experiments, this is the case.

To backup our analysis, we tested the scalability of our algorithm on a data tensor of English 3-grams. We varied the number of non-zeros in the data tensor from five million down to a few hundred by removing non-zeroes uniformly at random. We used a laptop with 8GB of memory and 1.3GHz of CPU to run Algorithm 1 on these tensors with max-size = 100, min-size = 5, and $\phi^* = 0.4$. Figure 1 shows the results, and we see that the scaling is roughly linear.

Figure 1: Time to compute a partition on the English 3-grams as a function of the number of non-zeros in the tensors. We see that the algorithm scales roughly linearly in the number of non-zeros (the red line).

## C  Experiment discussion

### C.1  Clustering methods and evaluation for synthetic experiment

We compared the results of our GTSC framework to several other state-of-the-art methods for clustering tensor data.

**GTSC.** This is the method presented in this paper (Algorithm 1). We use the parameters $\alpha = 0.8$, `max-size` $= 100$, `min-size` $= 5$, and $\phi^* = 0.35$.[1]

**TSC.** This is the original tensor spectral clustering algorithm [3]. We use the algorithm with recursive bisection to find 20 clusters.

**PARAFAC.** The PARAFAC method is a widely used tensor decomposition procedure [7] that finds an approximation to the tensor by the sum of outer-products of vectors. We compute a rank-20 decomposition using the Tensor Toolbox [1, 5], and then assign nodes to clusters by taking the index of the vector with highest value in the nodes index. We use the default tolerance of $10^{-4}$ and a maximum of 1000 iterations.

**Spectral Clustering (SC).** Our clustering framework (Algorithm 1) works on mode-2 tensors, i.e., matrices. In this case, with $\alpha = 1$, our algorithm reduces to a standard spectral clustering method. We create a matrix $\boldsymbol{M}$ from the tensor data $\underline{\boldsymbol{T}}$ by summing along the third mode: $M_{ij} = \sum_{k=1}^{\bar{n}} \underline{T}_{i,j,k}$. We then run Algorithm 1 with the same parameters as GTSC.

**Multilinear Decomposition (MulDec).** This is the higher-way co-clustering method for tensor decomposition [10]. The parameter $\lambda$ is set to be 0. We compute the rank-20 decomposition and recover the clusters.

**Evaluation metrics.** We evaluate the clustering results using the Adjusted Rand Index (ARI) [8], Normalized Mutual Information (NMI) [9], and F1 score. The ground truth labels correspond to the generated groups.

### C.2  Real-world dataset

In all of our experiments, we use the stopping criterion $\alpha = 0.8$, $\phi^* = 0.4$, `max-size` $= 100$ and `min-size` $= 5$ for Algorithm 1.

Table 1 shows the airline-airport co-clusters our GTSC finds. The other two methods (i.e., SC and TSC) cannot find the *Worldwide metropolises* group, which is analogous to the "stop word" group in the n-gram experiments. In particular TSC Even has trouble accurately identifying the regional

Table 1: High-level descriptions of the larger co-clusters found by our GTSC framework on the Airline-airport dataset. The algorithm finds one co-cluster of international hubs and large commercial airlines and several geographically coherent groups.

| Name | # Airports | # Airlines | Airports description | Airlines description |
|---|---|---|---|---|
| Worldwide metropolises | 250 | 77 | Large hubs, e.g., Beijing Capital and JFK in New York | Large commercial airlines, e.g., United, Air Canada, Air China |
| Europe | 184 | 32 | 177 in Europe, rest in Morocco | 29 European airlines |
| United States | 137 | 9 | 136 in U.S., Cancún International | 29 all U.S. airlines |
| China/Taiwan | 170 | 33 | 136 in China or Taiwan, | 21 in China/Taiwan 14 in S. Korea and Thailand |
| Oceania/S.E. Asia | 302 | 52 | 231 in Oceania or S.E. Asia, | 41 in East Asia or Canada 66 in China, Japan, or Canada |
| Mexico/Americas | 399 | 68 | 396 in Mexico or Central and South America | 43 in Mexico or Central and South America |

Figure 2: Enron email volume on three labeled topics. Our GTSC framework finds a co-cluster consisting of these three topics at the time points labeled in red, which seems to correlate with various events involving the CEO.

clusters. Other methods like PARAFAC and MulDec cannot handle the dataset of this size. Although the number of non-zeros in this dataset is only $51,982$, PARAFAC and MulDec are unable to utilize the sparse structure of the tensor. As a result the data size is $4,655,731,619$ (i.e., $539 \times 2939 \times 2939$) to them.

## C.3 Extra experiemt - Enron email tensor

Due to space limit of the paper, we put our experiment results for Enron dataset here in Appendix.

**Enron email.** This dataset is constructed from emails between Enron employees with labeled topics [4]. The tensor data represents the volume of communication between two employees discussing a given topic during a particular week. In total, there are 185 weeks of data, 184 employees, and 34 topics, leading to a $185 \times 184 \times 184 \times 34$ tensor where $\underline{T}_{ijkl}$ is the number of emails between employee $j$ and $k$ on topic $l$ during week $i$.

In total, the algorithm finds 23 co-clusters of topics, people, and time. The most popular group corresponds to three topics, 19 people, and 0 time intervals. Similar to the $n$-grams and airport-airline data, this cluster corresponds to high-volume entities, in this case common topics and people who send a lot of emails. The three topics are "Daily business", "too few words", and "no matching topic",

which account for roughly 90% of the total email volume. (The latter two topics are capturing outliers and emails that do not fall under an obvious category). The 19 employees include 11 managers: the CEO, (vice) preseidents, and directors. These employees are involved in 42% of the total emails. There is no time interval in this co-cluster because these high-volume topics and employees are balanced throughout time.

We also found several interesting co-clusters. One consists of the topics "California bankruptcy", "California legislature", and "India (general)", during three weeks in December 2000 and January 2001, and 13 employees. These time points correspond to various events involving CEO Skilling (Figure 2). Each of the 13 employees in the co-cluster sent at least one email from at least one of the topics. Another co-cluster consists of the topics "General newsfeed", "Downfall newsfeed", and "Federal Energy Regulatory Commission/Department of Energy" and several weeks from March 2001 and December 2001. These time intervals coincide with investor James Chanos finding problems with Enron in early 2001 and the serious financial troubles encountered by the company in late 2001.

## D  Other discussion

### D.1  Dangling correction effect

The TSC framework [3] used a pre-defined stochastic dangling vector $\mathbf{u}$ when encountering a zero column of the tensor $\underline{\mathbf{P}}$. So the transition tensor is

$$\underline{\tilde{P}}_{ijk} = \begin{cases} \underline{P}_{ijk} & \text{if } \sum_{i=1}^{n} \underline{T}_{ijk} > 0 \\ u_i & \text{otherwise} \end{cases}$$

Denote $\tilde{\mathbf{R}}$ and $\mathbf{R}$ the mode-1 unfolding of tensor $\underline{\tilde{\mathbf{P}}}$ and $\underline{\mathbf{P}}$ respectively. Then we have

$$\tilde{\mathbf{R}} = \mathbf{R} + \mathbf{u}(\mathbf{e}^T - \mathbf{e}^T \mathbf{R}).$$

The multilinear Pagerank vector $\mathbf{x}$ of $\tilde{\mathbf{R}}$ from Equation (5) satisfies:

$$\begin{aligned} \mathbf{x} &= \alpha\big(\mathbf{R} + \mathbf{u}(\mathbf{e}^T - \mathbf{e}^T \mathbf{R})\big)\mathbf{x} \otimes \mathbf{x} + (1-\alpha)\mathbf{v} \\ &= \alpha \mathbf{R}\mathbf{x} \otimes \mathbf{x} + \alpha \mathbf{u}(1 - \mathbf{e}^T \mathbf{R}\mathbf{x} \otimes \mathbf{x} + (1-\alpha)\mathbf{v} \\ &= \alpha \mathbf{R}\mathbf{x} \otimes \mathbf{x} + \alpha(1 - \|\mathbf{R}\mathbf{x} \otimes \mathbf{x}\|_1)\mathbf{u} + (1-\alpha)\mathbf{v} \end{aligned}$$

where $\|\cdot\|_1$ denotes the $1-$norm of a vector.

When the tensor is very sparse, there are lots of zero columns in $\mathbf{R}$, and $\|\mathbf{R}\mathbf{x} \otimes \mathbf{x}\|_1$ can be quite small. Thus, the contribution of the dangling vector $\mathbf{u}$ becomes significant. In prior work, $\mathbf{u}$ is simply chosen as the uniform vector, and thus it washes out information about the structure in $\mathbf{R}$.

With the dangling vector, the spacey random surfer ends up guessing some history steps that are not feasible, i.e., states corresponding to zero columns in $\mathbf{R}$.

## Footnotes

[1] We tested several values $\phi^* \in [0.3, 0.4]$ and obtained roughly the same results.