[Reviews · NeurIPS 2016]

Reviewer 1

Summary

The paper aims to introduce a method for spectrally co-clustering tensor data. The proposed approach is based on a probabilistic considerations as opposed to factorization based approaches.

Qualitative Assessment

This is a nicely written paper. The paper succeeds in packing a lot of ideas and details in 8 pages at the expense of some imprecisions. Some comments about aspects of the presentation that could be improved: -- the writing is sometimes ambivalent around mathematical formulas. At times it is not clear whether the authors are making an important definition or just mentioning in passing a possible interpretation. A suggestion in this respect is that authors adopt a more authoritative tone when defining mathematical concepts. -- notation/terms are sometimes undefined. Notes: Line 83: please qualify $(\sigma_1,\ldots,\sigma_n)$. A boldface \sigma appears in Line 235. Line 91: we generalizeS -- remove "s" Lines 58, 185, 188: what is \mathbf{e}? Line 133: define Ind Line 119: Please provide references for "higher order Markov Chains" Line 170. I assumed you are assuming from that line onwards that alpha satisfies the conditions of Theorem 2.1. It would help stating this explicitly.

Confidence in this Review

2-Confident (read it all; understood it all reasonably well)


Reviewer 2

Summary

This paper proposes a General Tensor Spectral Co-clustering (GTSC) method for analyzing higher-order tensor data. The problem is well motivated. The paper is well written and it is delightful to read through it, especially the first section. The method is built on several existing works with some novel extensions so there could be some concern regarding the novelty and claims. The major weakness, however, is the experiment on real data where no comparison against any other method is provided. Please see the details comments below.

Qualitative Assessment

1. While [5] is a closely related work, it is not cited or discussed at all in Section 1. I think proper credit should be given to [5] in Sec. 1 since the spacey random walk was proposed there. The difference between the random walk model in this paper and that in [5] should also be clearly stated to clarify the contributions. 2. The AAAI15 paper titled "Spectral Clustering Using Multilinear SVD: Analysis, Approximations and Applications" by Ghoshdastidar and Dukkipati seems to be a related work missed by the authors. This AAAI15 paper deals with hypergraph data with tensors as well so it should be discussed and compared against to provide a better understanding of the state-of-the-art. 3. This work combines ideas from [4], [5], and [14] so it is very important to clearly state the relationships and differences with these earlier works. 4. End of Sec. 2., there are two important parameters/thresholds to set. One is the minimum cluster size and the other is the conductance threshold. However, the experimental section (Sec. 3) did not mention or discuss how these parameters are set and how sensitive the performance is with respect to these parameters. 5. Sec. 3.2 and Sec. 3.3: The real data experiments study only the proposed method and there is no comparison against any existing method on real data. Furthermore, there is only some qualitative analysis/discussion on the real data results. Adding some quantitative studies will be more helpful to the readers and researchers in this area. 6. Possible Typo? Line 131: "wants to transition".

Confidence in this Review

2-Confident (read it all; understood it all reasonably well)


Reviewer 3

Summary

This paper proposes new tensor spectral co-clustering method based on a new random walk model. The proposed method generalized Tensor Spectral Clustering (TSC) algorithm [4] to handle the issue of undefined transitions via a new super-spacey random surfer. Extensive experiments indicate the superior performance of the proposed method.

Qualitative Assessment

Q1: Can the authors discuss the case when \alpha \ge 1/(2m-1) in Theorem 2.1? In addition, since \alpha is unknown in practice, how to choose \alpha in a data-driven manner? Q2: In Algorithm in Section 2.6, the authors mentioned that ``We continue partitioning as long as the clusters are large enough or we can get good enough splits". However, in practice, how to choose cluster size? how to choose the target value of biased conductance? Q3: In the experiments. e.g. Table 1, the authors need to explain in details how the competitive methods are implemented. For example, which tuning parameters (e.g., rank K) are used in these methods? How Tensor Decomposition PARAFAC method is used for co-clustering?

Confidence in this Review

2-Confident (read it all; understood it all reasonably well)


Reviewer 4

Summary

This paper develops a tensor spectral clustering method. It is based on probability transition tensor, describing a 2nd or higher order Markov process. The method works by finding the stationary distribution of the process, on which conventional spectral clustering can be applied. The major contribution is a "better" definition of transition probability when there are infeasible states.

Qualitative Assessment

Section 2.2 presents the key framework --- getting stationary distribution matrix and its eigen-decomposition. Yet it is not well presented. As a reader who knows conventional spectral clustering well, I don't have any intuition why it works. The authors should make it more friendly to the readers. The major contribution is Eq. (8), a definition of transition probability when the previous state is infeasible. Again I cannot see any reason behind the definition. The authors should make it well motivated. The experiments on two real datasets are interesting, but not well analyzed. It is hard to judge the advantage of the proposed method on these data.

Confidence in this Review

1-Less confident (might not have understood significant parts)


Reviewer 5

Summary

This paper propose a new variant of the super-spacey random walk - a variant of higher-order Markov chain model that has been introduced in previous work for co-clustering tensor data. Experimental results show that the algorithm outperforms several previous works on synthetic dataset. The algorithm is then evaluated on several realistic datasets.

Qualitative Assessment

I'm not an expert in the problem of clustering but I have read the proofs in some detail including the supplementary material and did not notice significant errors. Overall, I find the paper well-written and understandable. The supplement is well-prepared and contains more details about experimental results which I appreciate. In terms of technical contributions, the problem of undefined transitions in the probability tensor has been tackled already in previous work [4]. Here the authors consider a different way to solve it. However, since the two models share striking similarities, I think the new model should be justified more in terms of both theoretical motivation and analytical results (see below) so that one can clearly see the advantages/drawbacks of the model compared to previous work. As I'm not an expert in this problem, I would consider most of my comments below as suggestions for improving the paper if it is accepted. 1. Please check carefully the notations & variable's indices between lines 44-45 in the supplement as I believe there are some mistakes. 2. Is there any equivalence form of the newly introduced biased conductance like the standard one as shown in Observation 1. At this point, paper [4] has proposed similar conductance functions - "higher-order conductance" and "higher-order expansion". Please comment on this. 3. The biased conductance has been introduced in the paper as a new quality metric for partitioning data. However, it is never used in the experiments to analyze the algorithm, but instead other evaluation metrics (ARI,FMI,F1) were used to compare all algorithms. Some figures showing the values of different higher-order conductance as functions of |S_k| for different methods (at least GTSC,TSC,SC) could be helpful to readers. 4. It seems that the comparison with previous work is done only on synthetic datasets with quantitative results, while it is rather qualitative on other real-world datasets. In this way, I cannot see why one should use the proposed algorithm in favour of other methods for practical clustering. Moreover, on the synthetic dataset, only 5 trials that has been used to estimate the standard deviations for algorithms might be too small so that one cannot say much about standard deviations.

Confidence in this Review

2-Confident (read it all; understood it all reasonably well)


Reviewer 6

Summary

This paper develops a General Tensor Spectral Co-clustering (GTSC) method to cluster non-negative, high-order tensor data which may be sparse, non-square and asymmetric. It is an extension of the existing Tensor Spectral Clustering (TSC) method. It generalizes the spacey random surfer stochastic process to a super-spacey random surfer that picks a random state from history when the transitions are not defined. It forms a higher-order Markov chain from non-negative tensor data and derives an equivalent first-order Markov chain from the stationary distribution of the super-spacey random surfer. Then it introduces a biased conductance measure and partitions the non-reversible Markov chain by the sweep cut procedure of spectral clustering.

Qualitative Assessment

This paper develops a GTSC method to cluster high-order tensor data which may be sparse, non-square and asymmetric. The explanations are clear, and the results are reasonable although not very strong. It can be improved in the following ways: 1. elaborate on the efficiency, as tensor data is likely to be large-scale 2. discuss the connection/distinction between (super)-spacey stochastic process and partially absorbing random walks (Learning with partially absorbing random walks, nips'12) 3. show one compelling application that shows the necessity of using tensor data and any efforts in flattening the tensor (into lower-order or even matrix) could lead to significant loss

Confidence in this Review

2-Confident (read it all; understood it all reasonably well)